

# A novel multi proxy approach reveals that the millennial old ice cap on Weißseespitze, Eastern Alps, has preserved its chemical and isotopic signatures despite ongoing ice loss

**Azzurra Spagnesi**[1,2], **Pascal Bohleber**[1,3], **Elena Barbaro**[2], **Matteo Feltracco**[1], **Fabrizio De Blasi**[1,2], **Giuliano Dreossi**[1,2], **Martin Stocker-Waldhuber**[3], **Daniela Festi**[4], **Jacopo Gabrieli**[2], **Andrea Gambaro**[1,2], **Andrea Fischer**[3], **Carlo Barbante**[1,2]

[1] Department of Environmental Sciences, Informatics and Statistics, Ca' Foscari University of Venice, Venice, Italy

[2] CNR-Institute of Polar Sciences (ISP-CNR), 155 Via Torino, 30170 Mestre, Italy

[3] Institute for Interdisciplinary Mountain Research of the Austrian Academy of Sciences, Innrain 25/3, 6020 Innsbruck, Austria

[4] GeoSphere Austria, Neulinggasse 38, 1030 Vienna, Austria

**Correspondence:** Azzurra Spagnesi (azzurra.spagnesi@unive.it)

## Abstract

From the 1970s to the early 2000s, Alpine ice core research focused on a few suitable drilling sites at high elevation in the Western European Alps, assuming that the counterparts at lower elevation in the eastern sector are unsuitable for paleoenvironmental studies, due to the presence of melting and temperate basal conditions. Since then, it has been demonstrated that even in the Eastern Alpine range, below 4000 m a.s.l., cold ice frozen to bedrock can exist. In fact, millennial-old ice has been found at some locations, such as at the Weißseespitze (WSS) summit ice cap (Ötztal Alps, 3499 m a.s.l.), where about 6 kyrs appear locked into 10 m of ice. In this work, we present a full profile of the stable water isotopes ($\delta^{18}O$, $\delta^2H$), major ions ($Na^+$, $Cl^-$, $Br^-$, $K^+$, $Mg^{2+}$, $Ca^{2+}$, $NO_3^{2-}$, $SO_4^{2-}$, $NH_4^+$, $MSA^-$), levoglucosan, and microcharcoal for two parallel ice cores drilled at the Weißseespitze cap. We find that, despite the ongoing ice loss, the chemical and isotopic signatures appear preserved, and may potentially offer an untapped climatic record. This is especially noteworthy considering that chemical signals of other archives at similar locations have been partially or full corrupted by meltwater (i.e., Silvretta glacier, Grand Combin glacier, Ortles glacier). In addition, the impurity concentration near the surface shows no signs of anthropogenic contamination at WSS, which constrains the age at the surface to falls within the pre-industrial age.

## 1. Introduction

European Alpine glaciers represent unique targets for ice core studies focusing on reconstructing environmental and climatic conditions in the Holocene. Since its beginning, a primary aim of ice core research in the Alps was retrieving continuous stratigraphic climate records, which restricted it to glaciers without significant melting on the surface throughout the year. In this strict view, only few suitable drilling sites exist as they are mostly confined to above 4000 m altitude and hence located in the Western Alps, which have been exploited in numerous successful studies over the past four decades (Bohleber, 2019, and references therein). Following evidence that old ice may also exist in the Eastern Alps and at elevations below 4000 m (Haeberli et al., 2004), new efforts targeted the either direct access to the ice at the glacier base (Bohleber et al., 2018) or the drilling of ice cores at both temperate (Pavlova et al., 2015; Festi et al., 2017), partially temperate but





cold-based (Gabrielli et al., 2016) and predominantly cold ice sites (Bohleber et al., 2020a). In concert with state-of-the-art radiocarbon ice dating (Uglietti et al., 2016; Hoffmann et al., 2018), the access to the stagnant cold ice at the glacier base revealed that millennial-old ice is still preserved even if the remaining thickness is only around 10 m or less, adding important information for reconstructing the Holocene neoglaciation history of the Alps (Bohleber et al., 2020a). Retrieving the paleoclimate and environmental information potentially stored in these ice cores' chemical and isotopic stratigraphy means facing additional complexity to what it is already known for the "classical" ice core targets in the Western Alps (Wagenbach et al., 2012). First, due to ongoing and prolonged mass loss also the age of the ice at the surface becomes an unknown parameter that requires separate dating efforts with innovative approaches (Festi et al., 2021). Second, due to their lower elevation, these sites are typically much closer to the equilibrium line altitude of the glacier, making them not only more vulnerable to present warming conditions but also a potential sensitive indicator of past climate shifts impacting their energy and mass balance. In fact, prolonged periods of stagnation or mass loss may have occurred and resulted in stratigraphic discontinuities at such sites (Fischer et al., 2022). Third, meltwater percolation can corrupt and ultimately erase completely the chemical and isotopic information in the stratigraphy, although the degree of the disturbances may depend on the impurity species and the degree of percolation (Avak et al., 2018; Eichler et al., 2001).

Here we present recent progress in evaluating these challenges for the new ice cores drilled at the Weißseespitze cap, Eastern Alps (WSS). We use for this purpose profiles of the stable water isotopes ($\delta^2$H and $\delta^{18}$O), major ion chemistry as well as a full profile of microcharcoal and levoglucosan. The latter represents a novelty for ice core studies over the alpine range, since only a preliminary work at Col du Dome is currently available (Legrand et al., 2007). Levoglucosan was measured on the WSS ice core to investigate if evidence of past biomass burning events could be detected, since levoglucosan is particularly useful when source and deposition sites are close to each other (Alves et al., 2017).

## 2. Methods

### 2.1 Glaciological settings of the ice core drilling site at Weißseespitze

The Weißseespitze ice cap (around 3500 m a.s.l.) covers the top sections of Gepatschferner glacier in the Austrian Alps (Fig. 1a,b), located only 12 km from the famous Tyrolean Iceman find site. Its limited ice thickness combined with a dome-shaped glacier geometry entails minimal to no ice flow, confirmed by differential GPS measurements at stakes in 2018 and 2019. Historical photographs dating back to about 1888, maps and digital elevation models reveal that the ice body today is the remnant of a much larger ice cap diminished by prolonged ice loss (Fig. 1c,d). Despite the prolonged ablation, englacial borehole temperatures remained permanently sub-zero at 1 m below surface, with -3°C at 9 m of depth (Fischer et al., 2022). A first ice core was drilled to bedrock (11 m in total, including about 1 m of snow cover but no firn) at the ice divide with nearly flat bed conditions in March 2019. An additional parallel core (8.7 m of depth to bedrock) was drilled at the same location in March 2021. The few visible layers of refrozen meltwater in the cores indicate



that there was only limited occasional melt at this site when the ice formed. The main part of the ice cores
includes bubble-rich glacier ice, the likely result of dry metamorphosis of snow (Fig. 1e). Initial analysis
supported this view, e.g., by stable oxygen and hydrogen ratios, exhibiting a range typical for the seasonal
variation in snow at this altitude, and no systematic deviation from the meteoric water line (Bohleber et al.,
2020a). The 2019 surface at WSS is older than 1963, indicated by the absence of elevated tritium levels within
the first 4 m of the core. The aerosol-based micro-[14]C dating indicated a maximum age of (5.884 ± 0.739) ka
cal BP just above the bed. Further details on the glaciological settings have already been described in former
studies (Bohleber et al., 2020a; Fischer et al., 2022).

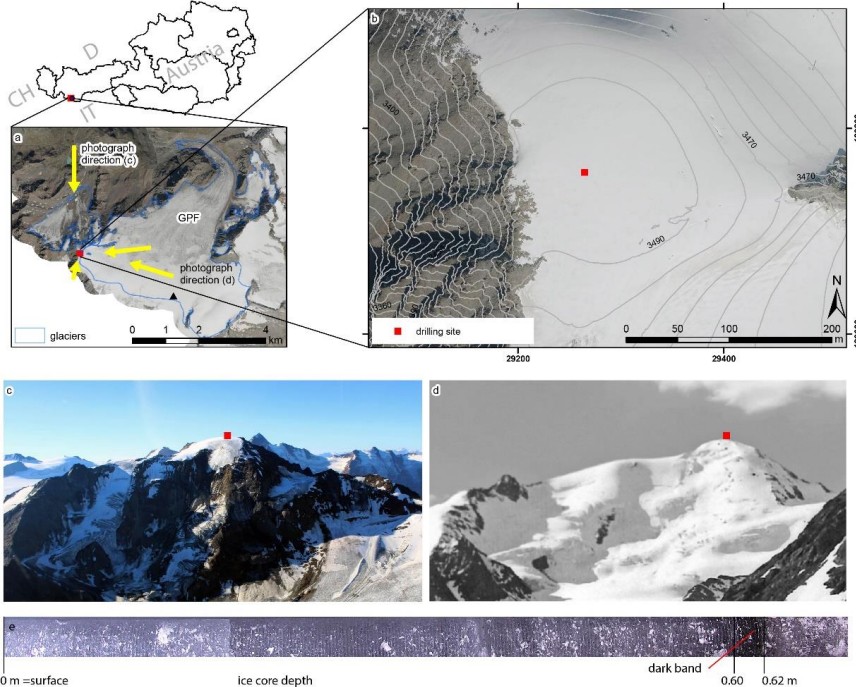

**Figure 1**. The ice core drilling site at the summit of the Weißseespitze ice cap (a, b). Datenquelle: Land Tirol
- data.tirol.gv.at. Panoramic view of the Weißseespitze ice cap in 2019 (c), and 1930 (d). One of the ice cores
taken in 2019 (e).
**2.2 Ice core processing and analysis**
To prepare for Continuous Flow Analysis (CFA) at Ca'Foscari University of Venice, the 2019 core was
processed to obtain 23 ice sticks (*bags*) with 32 x 32 mm sections. Only the top 8.5 m were considered suitable
for the analysis, given the high concentration of visible debris at the bottom. The ice was cut with a modified
commercial band saw, and refined with a decontaminated stainless-steel blade over a polyethylene tabletop
accessorized with guide rails for cutting. The table, rails, and the blade were carefully cleaned with acetone



and methanol to remove contamination before every use. All the exposed ice surfaces were rapidly scraped
with a stainless-steel knife cleaned with 0.1 % ultra-pure $HNO_3$ (Romil, Cambridge, UK). This knife was used
to remove the outer thin contaminated ice layer, and more mass was scraped from the two base surfaces which
were to be placed on the melting head. Several mm of ice from each end were removed by using a second
clean knife to ensure perfect contact to the melting head surface. The sections were stored in clean PTFE bags
until the analyses conducted with the novel set-up of the Continuous Flow Analysis system realized at Ca'
Foscari, in collaboration with the National Research Council - Institute of Polar Science (CNR-ISP). This
technique allows to continuously measure insoluble dust particles (1 acquisition $sec^{-1}$) and levoglucosan (1 cm
of resolution) within the meltwater stream, while sets of discrete samples (2.6 cm of ice depth equivalent per
sample) were reserved for the off-line analysis of water stable isotopes and major ions, conducted via Cavity
Ring-Down Spectrocopy (CRDS, Picarro inc.), and Ion Chromatrography (IC), respectively. The overall CFA
system coupled with Fast Liquid Chromatography tandem Mass Spectrometry (FLC – MS/MS), is illustrated
in Barbaro et al. (2022), while the optimization of the method for levoglucosan continuous measurements is
presented in Spagnesi et al. (2023).
In order to investigate the localization of the impurities in the ice matrix and their potential removal through
meltwater percolation under temperate conditions, exemplary sections were analysed at the University of
Venice by 2D chemical imaging with laser ablation inductively-coupled plasma mass spectrometry (LA-ICP-
MS). The LA-ICP-MS set-up comprised an Analyte Excite ArF excimer 193 nm laser (Teledyne CETAC
Photon Machines) and an iCAP-RQ quadrupole ICP-MS (Thermo Scientific), connected via a rapid aerosol
transfer line for fast washout. Samples surfaces are decontaminated with ceramic $ZrO_2$ blades (American
Cutting Edge, USA), and the sample is then placed on a cryogenic sample holder. A glycol-water mixture (-
35°C) is used to cool the sample surface to -23 +/-2°C which is further cleaned by preablation with an 80 x 80
μm square spot before each measurement. Further details are described in Bohleber et al. (2020)b.
Sample stripes of about 8 x 2 x 1 cm were cut from bags 2 and 18, at depths of 0.08 – 0.10 and 6.25 – 6.75 m,
respectively. Images were recorded using a 40 micron spot over areas that showed visual evidence of grains
and grain boundaries. Due to the comparatively large grains in bag 18, only one grain boundary was present
within the image. Analytes were Na, Mg, Al and Fe in order to consider species with mostly soluble (Na, Mg)
as well as insoluble (Al, Fe) behaviour.
The 8.4 m long ice core drilled in 2021 was cut in 106 continuous samples at 10 cm resolution from the surface
to 6.6 m of depth, and at 5 cm resolution from 6.6 m to the bottom. The ice was cut with a modified commercial
band saw. Samples were stored frozen in plastic bags and sent frozen to the Palynological Laboratory at Milano
Bicocca University for microfossils extraction (including microcharcoal) and preparation. Decontamination
was performed using fridge-cooled distilled water and left to melt covered at room temperature. Sample
volume was measured and samples were then filtered with a 7-μm filter to concentrate the microfossils (pollen,
spores, microcharcoal and other non-pollen-palynomorphs). The so concentrated samples underwent chemical
digestion according to Festi et al. (2019). Microscopy slides were prepared in the Milano laboratory and
analysed at the Institute for Interdisciplinary Mountain Research of the Austrian Academy of Sciences using



a Motic BA310 light microscope. Microcharcoal particles (> 7 µm) were quantified along with pollen grains
as usual in pollen analyses. For each sample, the complete content was analysed. In this work, we present the
microcharcoal record obtained for the core.
**3. Results**
**3.1 Water stable isotopes**

Water stable isotopes ($\delta^{18}$O, $\delta^2$H) and deuterium excess were measured in discrete samples, resulting in the
depth profiles shown in Fig. 2. The upper part of the core, ranging from 0 to roughly -2.30 m of depth, is
characterized by distinct decimeter-scale variability over several permil in $\delta^{18}$O. We find two marked minima
($\delta^{18}$O: -22 ‰, -20 ‰; $\delta^2$H: -165 ‰, -152 ‰) located at -0.63 m and -2.27 m of depth, respectively, with
isotopic signals similar to higher elevation Alpine sites (Wagenbach et al., 2012; Bohleber et al., 2013). By
comparison, the deeper part of the core below 6 m only shows minor variability around a stable mean (-14 ‰
± 1 ‰). However, there is clear decimeter-scale variability found over the entire depth range. Overall, the
deuterium excess record (d) does not show any clear trend, with values ranging between 10 and 15, similarly
to what observed by Fröehlich et al. (2008) for stations located north and south of the main ridge of the Austrian
Alps. Only one marked maximum, located around 4 m of depth, deviates from this general trend: this peak
corresponds to a minimum in $\delta^{18}$O and $\delta^2$H;other negatively-correlated values between d and $\delta^{18}$O ($\delta^2$H) can
be observed at -1.67, -4.99, -5.96 and -8.28 m of depth.
The slope in the co-isotopic plot is 7.76 ± 0.05, with a $R^2$ for the linear fit of 0.9864. This value is very close
with previous results obtained by Bohleber et al. (2020), for the same drill site, albeit at coarser depth
resolution. Notably, the slope reveals no systematic deviation from the local meteoric water line calculated on
the Villacher GNIP station monthly precipitation data between 1973 and 2002: $\delta D=8.09\delta^{18}O+12.48$ (IAEA,

162 2023).


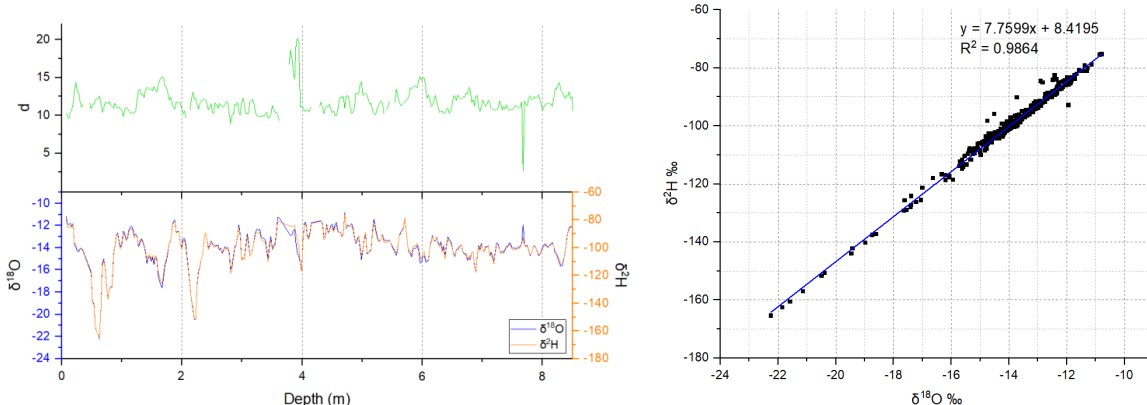

**Figure 2.** Deuterium excess (d), $\delta^{18}$O and $\delta^2$H profiles (‰) along the WSS ice core (left), and $\delta^{18}$O/$\delta^2$H linear
regression (right).



### 3.2 Major ions chemistry, levoglucosan and insoluble dust particles


The obtained levoglucosan profile shows concentration ranges between 0.07 and 51.07 ng mL$^{-1}$, with a major
peak found at ~ 6.40 m (Fig. 3). To investigate if this outstanding feature is connected to other proxy data, the
size-distribution of the insoluble dust particles was used to calculate a Fine Particle Percentage (FPP) along
the whole depth, summing the number of particles detected between 0.8 and 3.9 µm. The threshold of 3.9 µm
was chosen according to Wagenbach and Geis (1989), as an indicator for the median particle diameter of
Saharan dust, albeit at high Alpine locations. About 85.2 % ca. of particles are below 4 µm in size (ø < 4 µm),
almost equally distributed along the ice core, with a decrease of 4 % ca. for FPP observed for the deepest
section (from 6 m to 8.5 m).
Cationic (Na$^+$, NH$_4^+$, K$^+$, Mg$^{2+}$, Ca$^{2+}$), and anionic species (Cl$^-$, NO$_3^-$, SO$_4^{2-}$, Br$^-$, MSA$^-$) were analysed within
the discrete samples collected during the melting campaign. The mean concentrations, SD, minimum,
maximum, and median values of all the ionic compounds and water stable isotopes were computed over the
whole core depth and the upper 1 m separately, but showing no significant differences between the two sets
for NH$_4^+$, K$^+$, Ca$^{2+}$, NO$_3^-$, and Br$^-$(Table 1). The overall chemical composition is dominated by nitrate (NO$_3^-$,
37 %), sulphate (SO$_4^{2-}$, 23 %), calcium (Ca$^{2+}$, 12 %), and ammonium (NH$_4^+$, 11 %). Minor contributions were
reported for Cl$^-$, K$^+$, Na$^+$, and Mg$^{2+}$, respectively, quantified as 6.93 %, 3.82 %, 3.05 %, and 2.21 %, while
MSA$^-$ and Br$^-$ accounted for 0.58 % and 0.09 % of the total ionic species. The ionic balance was evaluated
considering the ionic concentrations in terms of equivalent, in order to evaluate the degree of neutralization.
The difference between the sum of anions and the sum of cations is always around ± 5 % over the whole profile
of the core. The prevalent cations/anions ratio > 1 indicates a limited defect of anions (Fig. S2), likely related
to the carbonates (CO$_3^{2-}$), which were not analysed in this work due to instrumental limits. Figure S2 displays
some values < 1, which can suggest the presence of cations deficit, likely due to the presence of H$^+$.
Regarding the localization of the impurities as investigated by LA-ICP-MS, Fig. 4 shows the near surface
sample (WSS bag 02), and the examples of Na, Mg and Al in a separate colour channel. All elements
(including Fe and Sr) are predominantly localized at grain boundaries, which are visible in Fig. 4 as lines of
bright intensity. The second sample from bag 18 in shown in the supplement, with the same basic finding (Fig.
S3).

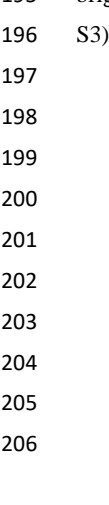
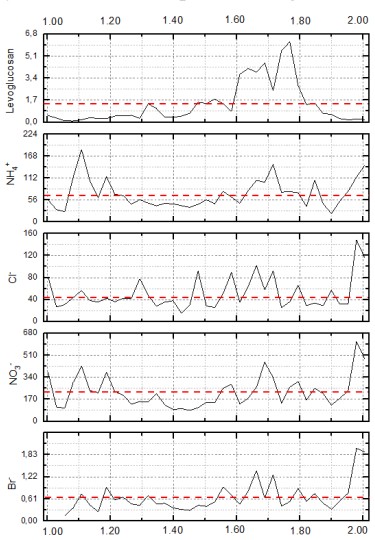
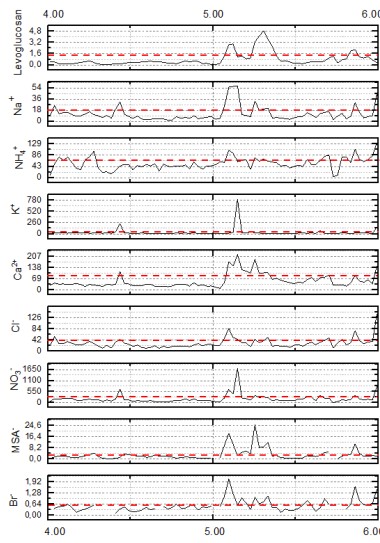




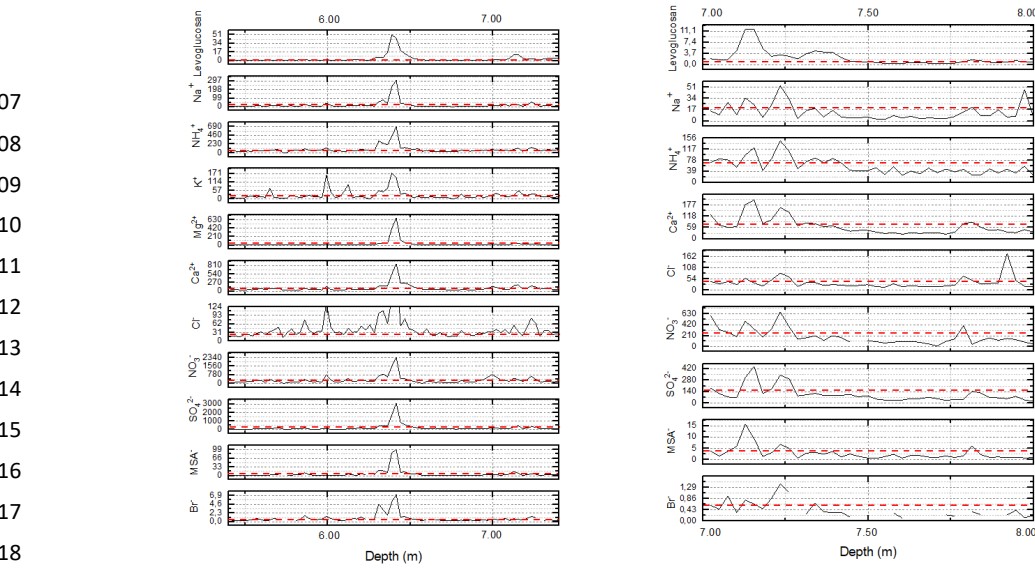

**Figure 3.** Zoom in from Fig. 4 at: (1) 1.60-1.90 m, (2) 5.00-5.50 m, (3) 6.40 m, and (4) 7.10-7.20 m of depth. Concentrations on y-axes are expressed in ng mL$^{-1}$. Red dashed lines indicate mean values.

**Table 1.** Ionic compound and water stable isotopes (WSI) average, min, max, and median values for the whole core depth and the upper 1 m (concentrations are expressed in ng mL$^{-1}$).

| Ionic compound and WSI | whole core | | | | upper 1 m | | | |
|---|---|---|---|---|---|---|---|---|
| | Average ± SD | Min | Max | Median | Average ± SD | Min | Max | Median |
| Na$^+$ | 18 ± 29 | 1.34 | 305 | 9.36 | 39 ± 40 | 4.90 | 166 | 20 |
| NH$_4^+$ | 67 ± 64 | 2.87 | 685 | 52 | 74 ± 78 | 12 | 440 | 56 |
| K$^+$ | 23 ± 52 | 1.59 | 802 | 13 | 30 ± 28 | 2.38 | 102 | 17 |
| Mg$^{2+}$ | 14 ± 45 | 1.85 | 673 | 6.60 | 5.42 ± 1.43 | 2.11 | 8.72 | 5.28 |
| Ca$^{2+}$ | 73 ± 74 | 11 | 866 | 53 | 75 ± 27 | 26 | 150 | 69 |
| Cl$^-$ | 42 ± 37 | 11 | 291 | 30 | 79 ± 62 | 21 | 248 | 46 |
| NO$_3^-$ | 226 ± 245 | 6.67 | 2371 | 154 | 204 ± 95 | 89 | 426 | 190 |
| SO$_4^{2-}$ | 144 ± 266 | 7.09 | 3119 | 79 | 97 ± 80 | 32 | 363 | 68 |
| MSA$^-$ | 3.73 ± 8.39 | 0.15 | 99 | 1.94 | 0.91 ± 0.69 | 0.15 | 2.21 | 0.59 |
| Br$^-$ | 0.62 ± 0.66 | 0.07 | 7.11 | 0.47 | 0.49 ± 0.25 | 0.13 | 1.07 | 0.47 |
| δ$^{18}$O ‰ | -14 ± 1 | -15 | -13 | -14 | -15 ± 3 | -18 | -12 | -14 |
| δ$^2$H ‰ | -100 ± 13 | -113 | -87 | -99 | -112 ± 24 | -136 | -88 | -102 |



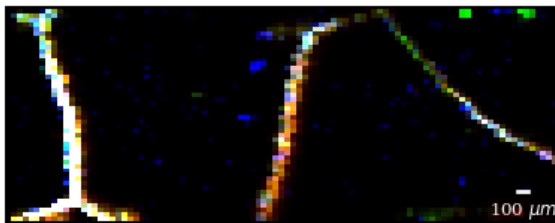


**Figure 4.** WSS ice core sample from bag 2. The LA-ICP-MS image consists of 50 lines measured with a 40 micron spot and shows Na, Mg, Al in red, green and blue color scale, respectively for an area of 2 x 5 mm. All elements are found mostly co-localized at grain boundaries (bright lines), while Al and Mg also form some isolated spots.

### 3.3 Microcharcoal

Microcharcoal concentration in the core ranges from 0 to 26.4 particles per mL (Fig. 5). Only 23 over 106 samples contained no microcharcoal and in the 8.4 m long profile, several peaks can be identified. The most prominent one is found 8 cm below the surface of the core, from 8 to 28 cm of depth and reaches the maximum concentration of 26.4 particles mL$^{-1}$. Applying a smoothing, three further composite peaks stand out at 3.5-4 m, 4.7-5.2 m and 5.6-6.2 m of depth.

## 4.    Discussion

### 4.1 A preserved chemical and isotopic record

The fact that no deviation from the meteoric water line is indicated in the co-isotopic plot suggests the absence of substantial melting and refreezing processes on the cm-scale (Craig, 1961), thus constituting an analogue situation to what was previously observed for other cold-based alpine summit sites (Bohleber et al., 2018; Bohleber et al., 2020a). This is also consistent with the fact that no clear visual evidence of refrozen ice manifesting as transparent bubble-free layers was found in visual analysis of this ice core. Throughout the entire record, notably including the near-surface layers, we find distinct variability in the chemical and isotopic signals obtained from the ice core. We can further exclude substantial percolation of meltwater, as this would have likely led to the continuous removal of impurities by gradual washing out, hence reducing and eventually removing any variability (Pavlova et al., 2015; Schotterer et al., 2004). This is consistent with what we have found in the exemplary LA-ICP-MS maps, which show a typical degree of impurity localization at grain boundaries previously observed in cold polar ice conditions (Souchez and Jouzel, 1984). We thus conclude that, despite the intense ablation which is acting at the surface of the Weißseespitze summit ice cap today, the cold ice remains mostly impermeable to meltwater which must be running off along the snow/firn layers. This is backed by the sub-zero temperatures measured inside the boreholes (Bohleber et al., 2020a; Fischer et al., 2022). As a result of these conditions, the ice contains preserved isotopic and chemical signals observed along the ice core depth. This is not to be expected a-priori considering what was revealed for other Alpine sites, such as the nearby Ortles (Gabrielli et al., 2016), the Silvretta glacier (Pavlova et al., 2015; Steinlin et al.,



2015) or the Grand Combin glacier (Huber et al., 2022), where the archives have been partially lost, and flat
signals (i.e., Silvretta glacier) or depleted impurity concentrations (i.e., Grand Combin) have been recorded.
Because only the lower, older and thinned ice layers remain today at the Weißseespitze site, any seasonal
variations of the isotopic signal is not detectable at the present resolution. Regarding the outstanding peak in
the deuterium-excess profile at around 4 m of depth, we find no evidence of an instrumental origin, hence this
feature may be indicative of a change at the moisture source (Fröehlich et al., 2002). In particular, it can be
hypothesized that this indicates a period of exceptional recycling of continental moisture or moisture masses
formed over the Mediterranean basin (Fröehlich et al., 2008).

**4.2 Old ice at the surface and significance of the average impurity concentrations**
Apart from the fact that the impurity record appears overall undisturbed, it is noteworthy that the near-surface
layers show no distinct difference in concentration levels with respect to the rest of the core (Table 1). Within
the last 80-100 years, a distinct increase in most impurity species is observed in other Alpine ice cores due to
anthropogenic emissions, in particular for $NO_3^-$, $SO_4^{2-}$, and $NH_4^+$ (Bohleber, 2019; Wagenbaach et al., 2012;
Schwikowski et al., 1999a; Schwikowski et al., 1999b; Preunkert et al., 2003). This indicates that the current
surface at WSS is not only missing the $^3H$ bomb horizon conventionally associated with the year 1963 as
detected previously by Bohleber et al. (2020), but that the present surface is in fact significantly older than
1963 and falls within the pre-industrial time period.
Notably, the $SO_4^{2-}/Ca^{2+}$ ratio calculated for the upper 1 m of the core gives a mean molar ratio of 0.56. This
is comparable to the ratio of 0.59 computed by Wagenbach et al. (1996) for the pre-industrial (pre-1930)
period, which points towards a similar contribution of Saharan dust to the snow chemistry at WSS. Correlation
analysis using the non-parametrical Spearman correlation matrix reveals that all ions positively correlate with
each other (significance level 0.05, Fig. S1), which also holds for their trend variability. Considering that most
impurities feature a distinct seasonality at high alpine sites due to a seasonal contrast in vertical mixing strength
of the atmosphere (Preunkert et al., 2001), systematic changes in the seasonality of snow deposition is a prime
candidate to introduce such an apparent coupling among different ionic species with different emission sources
(Wagenbach et al., 2012).
A more detailed interpretation of the impurity record and comparison with other Alpine ice core records would
benefit greatly from constraining the seasonal bias in net snow deposition, which is to be expected at an
exposed summit site. Monitoring today's conditions at the surface is unsuitable for this purpose because it is
dominated by ablation all throughout the year. As a crude attempt at investigating potential seasonal biases in
snow deposition, we took a closer look at the average levels in $\delta^{18}O$, which contains a known clear seasonal
cycle in precipitation. The closest nearby meteorological weather station is located at Villacher Alpe (VA, Lat.
46.60 N, Long. 13.67 E, 2156 m a.s.l.), for which also data from the Global Network on Isotopes in
Precipitation (GNIP) are available for the 1973-2002 period (IAEA, 2023). The average seasonal cycle was
calculated for this time period and subsampled for all seasons. Results were corrected for the elevation
difference (using -0.09 ‰/100 m, as reported in Siegenthaler and Oeschger, 1980), obtaining a VA $\delta^{18}O$ fall



average value of -14.05‰, which is close to the average value over the entire WSS record, being -13.98 ‰
(Table S1). This tentatively indicates that snow preservation is neither restricted to the winter nor the summer,
but is likely a mix. However, this comparison suffers again from the fact that the two-time intervals do not
overlap due to the lack of the 1973-2002 time period in the WSS core. Acknowledging the present uncertainty
in the snow deposition bias, the impurity levels can indicate the general pre-industrial background in
precipitation for this region.

**4.3 Outstanding peaks in levoglucosan, microcharcoal and impurities**
In the core of 2019 levoglucosan and chemistry peaks are visible at 1.60-1.90 m, 5.00-5.50 m, 6.40 m, and
7.10-7.20 m of depth, respectively (grey bars in Fig. 5, and zoom-ins Fig. 3). The third event at 6.40 m showed
a visual correspondence with all the investigated ions. At the same time, four outstanding periods are also
present in the microcharcoal data analysed in the 2021 core. This core was drilled at about 10 m distance from
the 2019 core. Notably, the measured ice ablation at the surface shows a strong gradient on a very short distance
between the stakes placed at the drilling sites, with up to 110 cm of ablation within the period 2019-2021.
Considering these surface changes and that both levoglucosan and microcharcoal are proxies of biomass
burning, it appears plausible to match the near-surface microcharcoal peak in 2021 with the first levoglucosan
peak of the 2019 core. The striking correspondence of the four levoglucosan peaks with the four main
microcharcoal concentration peaks (Fig. 5) supports the hypothesis that at least four main fire activity phases
are recorded in the WSS ice record.

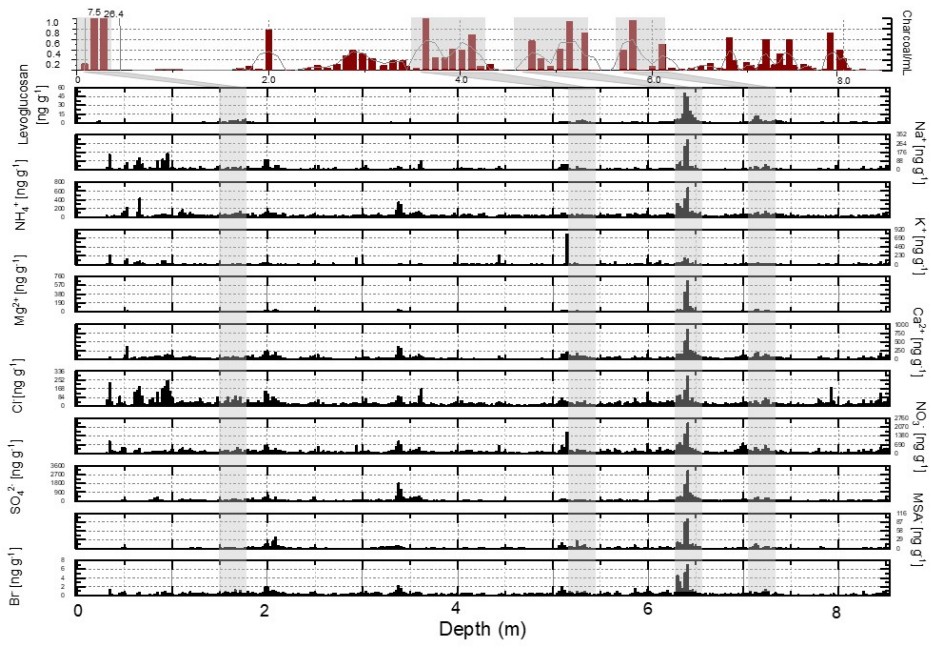




**Figure 5.** Full levoglucosan and chemistry profiles along the whole WSS longest core depth drilled in 2019.
Concentrations on y-axes are expressed in ng mL$^{-1}$. Grey bars indicate corresponding peaks between levoglucosan and
major ions. Microcharcoal/mL profile along the 8.4 m length WSS core drilled in 2021 is presented in the above graph
with brown bars.

### 5. Conclusion

The Weißseespitze summit ice cap has preserved the chemical and isotopic signature embedded in the ice,
despite the intense ice loss affecting the glacier's surface today. Indeed, the surface melting does not appear to
dominantly affect the remaining ice on the cm-scale, as indicated by a co-isotopic slope close to global
meteoric water line. The chemical signal in the upper meters of the ice does not show any evidence of the
anthropogenic increase during the 20$^{th}$ century known from other ice core. In particular, nitrates, sulphates,
and ammonium present pre-industrial concentrations. These results corroborate the previous age constraints
from $^3$H, showing the absence of the 1963 bomb horizon at the surface, but also indicate that today's surface
is much older and falls within the pre-industrial ear. The lack of temporal overlap between the WSS record
and instrumental data hampers constraining the seasonal representativeness and snow deposition bias so far.
However, four major peaks have been recognised standing out for levoglucosan but also other impurity species.
Based on the absence of evidence for disturbances by melting and refreezing, these events may stem from
either singular biomass burning events or a surface with prolonged exposure to the atmosphere during a hiatus.
With a future more robust estimate of the age-depth relation at WSS, comparisons with similar ice core alpine
records (e.g., ice cores from Alto dell'Ortles) or other natural archives (e.g., peatbogs), may offer new valuable
insights regarding the regional significance of these outstanding horizons.

### Data availability

Data will be made available on request.

### Author contributions

Conceptualization: Azzurra Spagnesi, Pascal Bohleber, Elena Barbaro; Methodology: Azzurra Spagnesi, Elena
Barbaro, Matteo Feltracco, Pascal Bohleber, Fabrizio De Blasi, Giuliano Dreossi, Jacopo Gabrieli; Formal
analysis and investigation: Azzurra Spagnesi, Elena Barbaro, Matteo Feltracco, Fabrizio De Blasi, Giuliano
Dreossi, Pascal Bohleber, Daniela Festi; Writing - original draft preparation: Azzurra Spagnesi, Pascal
Bohleber; Writing - review and editing: Azzurra Spagnesi, Pascal Bohleber, Andrea Fischer, Elena Barbaro,
Matteo Feltracco, Giuliano Dreossi, Daniela Festi, Martin Stocker-Waldhuber, Jacopo Gabrieli, Andrea
Gambaro, Carlo Barbante; Supervision: Jacopo Gabrieli, Andrea Fischer, Andrea Gambaro, Carlo Barbante.

**Competing interests:** The authors declare that they have no known competing financial interests or personal
relationships that could have appeared to influence the work reported in this paper.



**Acknowledgments**

This research was funded in part by the Austrian Science Fund (FWF) [I 5246-N and P34399-N]. For the purpose of open access, the author has applied a CC BY public copyright licence to any Author Accepted Manuscript version arising from this submission.

Pascal Bohleber gratefully acknowledges funding from the European Union's Horizon 2020 research and innovation program under the Marie Skłodowska-Curie grant agreement no. 101018266.

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
