# Peer review of "A novel multi proxy approach reveals that the millennial old ice cap on Weißseespitze, Eastern Alps, has preserved its chemical and isotopic signatures despite ongoing ice loss"

_EGUsphere, 2023_

## Referee Comment (RC2)

*Review of the manuscript entitled "A novel multi proxy approach reveals that the millennial old ice cap on Weißseespitze, Eastern Alps, has preserved its chemical and isotopic signatures despite ongoing ice loss" by Spagnesi et 11 co-authors*

The work reports on entire profiles of stable water isotopes ($\delta18O$, $\delta2H$), major ions ($Na^+$, $Cl^-$, $Br^-$, $K^+$, $Mg^{2+}$, $Ca^{2+}$, $NO_3^-$, $SO_4^{2-}$, $NH_4^+$, $MS^-$), levoglucosan, and microcharcoal from two parallel ice cores drilled at the Weißseespitze cap. From that the authors concluded that despite the ongoing ice loss, the chemical and isotopic signatures appear preserved, and may potentially offer an untapped climatic and environmental (including past biomass burning activity) record.

Overall evaluation: On several aspects of the manuscript, discussions appear to be unfinished. Discussions of the chemistry with previous studies of pre-industrial alpine ice is not done for many species. Details on methods are not given, especially for ion chromatography, hampering any evaluation of possible overlapping between Br- and NO3- for instance or between MSA and pyruvic acid, that are well known to occur with several separator columns. The scientific rational in discussing microcharcoal and levoglucosan is not correct. The evaluation of the acidic character of samples is inadequately conducted. The quality of some figures is very poor and misleading. Some basic references are missed. I was unable to identify a significant output of this work. For all these reasons, I consider that this material would be eventually publishable after major revisions as a brief communication, is absolutely not enough elaborated to be published as an article in TC.

Detailed comments:

**Title:** The title is inadequate and confusing: for me what is novel is the investigation of the ice of this old ice cap but in no way "a novel multiproxy approach". By the way what do you mean with "a multiproxy approach": proxies of what ? This point is never discussed in the text, except for levoglucosan used as a proxy of biomass burning (but see my further comment on that). Mg is a proxy of what ? MSA in the alpine ice to reconstruct what ? Br is a proxy of what, etc

**Abstract:**
Line 19: "from the 1970s to the early 2000s: please reword this sentence since there are numerous studies that have been recently conducted after 2000 in the 2010s and 2020s at the Mt Blanc summit (see the last one from Legrand et al., 2023 on bismuth).

Line 26: The abbreviate MSA has to be defined and anyway as it stands line 26 makes no sense: MSA abbreviates methanesulfonic acid that is not an anion. Please make your choice between MSA or MS-.

Line 26: Also, please note that nitrate is monovalent (not divalent!).

**Section 2.1:** Please report here the main findings from Bohleber et al., 2020 (your manuscript has to stand by itself).
The approach using $^{14}C$ to date the ice is not achieved with only one point at the bottom. In fact, since the authors know that this core did not archive the recent time it would have been

scientifically logic to proceed (even with growing age uncertainties) with measurement of $^{14}$C at different depth levels to estimate when the ice started to be preserved. On this point the work is clearly an unfinished work.

**Section 2.2 (analysis)**: There are no details on working conditions to evaluate the separation between Br and NO3 that can cause problem (see Legrand et al., JGR, 2016), also how are you sure that MSA is MSA and not pyruvate ?

**Section 3.2:** Discussion of the acidity or alkalinity. The approach used by the authors to identify acidic or non-acidic (alkaline) samples is not adequate (not very useful). Indeed, the authors used the cation to anion molar ratio (higher or lower than unit) to determine the acidic character but the derived information from that is very poor, being only qualitative without any estimate of the H$^+$ concentration. The authors can find in numerous previous papers how to calculate the H$^+$ (or HCO$_3^-$) concentrations (see for instance Preunkert and Legrand, CP, 2013).
The reason to investigate the localization of impurities here is not motivated. Furthermore, again here the discussion of data is quasi-inexistent: why there is some spots for Al and not for Na and Iron ? Is it a question of sensitivity of the method depending of species or a segregation between the two groups of species during the metamorphism of snow (grain versus grain boundary) (and if yes, why?).

**Section 4.2:**

Comparison with previous studies of pre-industrial alpine ice is limited to the sulfate to calcium ratio! That is an extremely reducing approach. Please read key studies previously conducted at the PSI, LGGE (now IGE), and IUP Heidelberg and compare your values with them. Since you show a bromine record please refer and compare to the unique (I think) previous study of Br in alpine ice (Legrand et al., JGR, 2022). As far as I know no previous studies reported MSA concentrations in alpine ice. Therefore, a discussion would have been welcome here.

Checking your data, I conclude that you definitely have no seasonal signal in this ice (as expected). Note at this point that your discussion on the variability of the vertical transport to explain the variability in your ice is totally incorrect. I am surprised by your high NO3 values compared to previous studies. Your Mg-Ca correlation is quite strange, with an outstanding ratio along your peak at 6.4 m depth (see below).

The discussion of the 6.4 m peak is not enough developed. For me it is definitely not a biomass burning event but a concentration of all species at a certain depth following a post-deposition process. To understand that more precisely I recommend to authors to check their data: for instance, I saw (see my previous comment) that the Mg to Ca ratio is totally different in these layers. That may help to understand the process of concern (see the papers from J. Moore in Svalbard ice for instance).

**Section 4.3:** This discussion of biomass burning records is not serious. Indeed, whereas it is right that microcharcoal can be used to reconstruct past biomass burning, it is less obvious for levoglucosan (since a lot of recent studies have highlighted that this species is less stable that previously thought). Furthermore, whereas we can expect that a microcharcoal peak may indicate the occurrence of a local (regional) fire that would also emit levoglucosan, the reverse is not true. Indeed, the levoglucosan stay in submicron organic aerosol particles whereas the

sizes of microcharcoal are far higher. So, an arrival of a long-range transported levoglucosan emitted by fires located far away from the Alps will not be detectable with microcharcoal.

I also find that the correspondence you propose between charcoal peak and levo levels not very convincing.

**Figures:** Some Figure are of poor quality, specially figures 3 and 5 that are messy pieces of work: For instance, in Figure 3 the vertical scale indicates a value of 6,9 ng g$^{-1}$ (for me that means 6900 ng g$^{-1}$). So , please change into 6.9 ng g$^{-1}$. Why the use of these strange numbers like 2760 ng g$^{-1}$ for nitrate for instance in figure 5 ?  Please revisit totally your figures 3 and 5 that will be rejected anyway at an editorial stage.

*End of the review*

Legrand, M., McConnell, J.R., Bergametti, G. *et al.* (2023). Alpine-ice record of bismuth pollution implies a major role of military use during World War II. *Scientific Reports*, **13**, 1166 (2023). https://doi.org/10.1038/s41598-023-28319-3

Legrand, M., McConnell, J. R., Preunkert, S., Chellman, N. J., & Arienzo, M. M. (2021). Causes of enhanced bromine levels in Alpine ice cores during the 20th century: Implications for bromine in the free European troposphere. *Journal of Geophysical Research: Atmospheres*, *126*, e2020JD034246. https://doi. org/10.1029/2020JD034246

Legrand, M., X. Yang, S. Preunkert, and N. Theys (2016), Year-round records of sea salt, gaseous, and particulate inorganic bromine in the atmospheric boundary layer at coastal (Dumont d'Urville) and central (Concordia) East Antarctic sites, *J. Geophys. Res. Atmos.*, 121, doi:10.1002/ 2015JD024066, 2016.

Preunkert, S., and M. Legrand, Towards a quasi-complete reconstruction of past atmospheric aerosol load and composition (organic and inorganic) over Europe since 1920 inferred from Alpine ice cores, *Clim. Past*, 9, 1403-1416, doi:10.5194/cp-9-1403-2013, 2013.